# Effects of Performing Applied Muscle Tension during Recovery after Phlebotomy in Young, First-Time Donors: A Pilot Study

**DOI:** 10.3390/ijerph181910541

**Published:** 2021-10-08

**Authors:** Cara H. Y. Cheung, May L. Khaw, Wan Shun Leung, Shing Yau Tam, Chui Yee Chu, Cheuk Kwong Lee, Shara W. Y. Lee

**Affiliations:** 1Department of Health Technology and Informatics, Faculty of Health and Social Sciences, The Hong Kong Polytechnic University, Hong Kong, China; carahy.cheung@polyu.edu.hk (C.H.Y.C.); wsv.leung@polyu.edu.hk (W.S.L.); marco-shing-yau.tam@polyu.edu.hk (S.Y.T.); 2Tasmanian School of Medicine, University of Tasmania, Hobart, TAS 7005, Australia; mlkhaw@utas.edu.au; 3Blood Collection and Donor Recruitment Department, Hong Kong Red Cross Blood Transfusion Service, HA, Hong Kong, China; ccy657@ha.org.hk (C.Y.C.); ckleea@ha.org.hk (C.K.L.)

**Keywords:** vasovagal reaction, haemodynamics, blood donation, applied muscle tension, young first-time donors, cardiac output, stroke volume

## Abstract

Vasovagal reaction (VVR) compromises donor safety and reduces the subsequent return rates. Performing applied muscle tension (AMT) during phlebotomy may reduce the incidence of VVR. However, the effectiveness of performing AMT after phlebotomy to reduce delayed VVR remains unclear. With ethics approval, 12 young, first-time donors (YFTD) were recruited to study the effects on stroke volume (SV), cardiac output (CO) and systemic vascular resistance (SVR) while performing AMT from needle insertion to end of recovery. Measurements from 12 matched control YFTD were used for comparison. Pre-donation anxiety and VVR severity were assessed. Compared to controls, donors who performed AMT had higher SV (Control: 57 mL vs. AMT: 69 mL, *p* = 0.045), higher CO (Control: 3.7 L·min^−1^ vs. AMT: 5.2 L·min^−1^, *p* = 0.006) and lower SVR (Control: 1962 dyn·s·cm^−5^ vs. AMT: 1569 dyn·s·cm^−5^, *p* = 0.032) during mid-phlebotomy. During recovery, the AMT group retained higher SV, higher CO and lower SVR than the control, but not reaching statistical significance. Practicing AMT during recovery resulted in sustained haemodynamic improvements beyond the donation period, despite the reduction in delayed VVR was insignificant compared to the control group. A larger sample size is needed to validate the effectiveness of performing AMT after donation to mitigate delayed VVR.

## 1. Introduction

The ability to keep donors safe and comfortable is an important prerequisite for a blood donation service. In Hong Kong, there is an urgent need for blood as blood donors in 2020 only represented 2.34% of the eligible population [1]. According to a local news report in 2020, the amount of donated blood is barely sufficient to sustain the daily demand in local hospitals [2]. According to the statistics recorded by the Hong Kong Red Cross Blood Transfusion Service (HKRCBTS), there is a decreasing trend in all categories including the total amount of blood donors, total units of blood collected and first-time donors from 2015 to 2018. One possible reason for such decline is due to the manifestation of blood donation related complications.

Vasovagal reaction (VVR) is a common complication of blood donation which may cause harm when severe. Donors who develop VVR are less likely to return to donate again. Our donor return rates dropped from 33.5% to 21.1% for first-time donors, and from 66.8% to 31.5% for repeat donors if they developed VVR [3]. The manifestations of VVR vary from mild symptoms such as dizziness, fatigue, weakness, nausea and vomiting, to falls or even syncope leading to head injuries [4,5]. Both physiological and psychological factors contribute to the aetiology of VVR. Physiologically, a reduction in the circulating blood volume compounded by orthosis may overcome compensatory mechanisms, resulting in haemodynamic compromise. The pathophysiological mechanism involves stimulation of the central nervous system leading to an increased vagal tone and a decreased peripheral vascular sympathetic nerve activity including skeletal muscles [6]. A high anxiety state and donor fear are important psychological factors, particularly in young, first-time donors (YFTD) during phlebotomy.

Applied muscle tension (AMT) is an effective manoeuvre that can be used to prevent VVR [7]. Although the exact mechanism is unclear, AMT has been reported to increase venous return and sympathetic tone while also distracting donor attention to reduce anxiety and fear [8,9,10], thereby mitigating both physiological and psychological causes of VVR. AMT for blood donation is usually performed during phlebotomy, starting from the point of needle insertion until removal. The usual procedure following completion of phlebotomy is to keep donors reclined in the donation chair to recover for 10 min before sitting up and mobilizing. However, several of the YFTD occasionally experienced VVR during the recovery period. To address this issue, we hypothesize that additional AMT performed prophylactically in the recovery period may reduce the incidence of VVR in the YFTD.

The aim of this pilot study was to evaluate the effects of performing an extended AMT during phlebotomy and recovery, on the haemodynamic indices and the incidence of VVR as measured by changes in the Blood Donation Reactions Inventory (BDRI) scores. The findings will facilitate the design of a large-scale study to evaluate if outcomes such as vasovagal syncope can be reduced after performing an extended AMT prophylactically in the recovery period.

## 2. Materials and Methods

This study received ethics committee approval from The Hong Kong Polytechnic University (Approval no.: HSEARS20181005001) and was compliant with the Declaration of Helsinki. It was conducted at the HKRCBTS and 12 (6 males and 6 females) healthy Chinese YFTD were recruited after giving full written informed consent. Participants were briefed on the study procedure and instructed on how to perform AMT. This consists of repeating 30 s tense–relax cycles of lower body muscle tension from buttocks to toes. This extended AMT was performed continuously from the time of needle insertion until the end of the 10 min recovery period. Donors were kept in the semi-recumbent position on the donation chair throughout the phlebotomy and recovery periods.

### 2.1. Data Collection

Pre-donation anxiety and fear levels were evaluated before phlebotomy using a standard two-sided State-Trait Anxiety Inventory (STAI) questionnaire with an additional fear assessment statement of “I am afraid of having blood being taken from my arm.” [11,12]. Five sets of haemodynamic measurements were collected using a vital signs monitor (Care Vision VS-110, Medical Supply Co., Ltd., Seoul, Korea) and an Ultrasonic Cardiac Output Monitor (USCOM 1A, USCOM PTY Ltd., Sydney, NSW, Australia) at: (1) baseline, (2) mid-donation (50% target volume), (3) end of donation (100% target volume), (4) 5 min post-donation and (5) 10 min post-donation. Donors were kept in the semi-recumbent position for all haemodynamic measurements. Vasovagal symptoms were assessed using a standard 11-point BDRI questionnaire after mobilization, and a second assessment for delayed VVR was made at 24 h after donation via text message. Donor experience of the AMT was evaluated using four questions: (1) did you perform AMT throughout the study? (Yes/No), (2) ease of performing AMT (3) amount of effort used (0–100%) and (4) willingness to use AMT in their next donation (0–100%). Prior to leaving, donors were verbally asked to rate their willingness to donate again. For this study, VVR was defined as ‘immediate’ if it occurred at the donation centre and ‘delayed’ if occurred off site and within 24 h of donation.

### 2.2. Study Design and Statistical Analysis

The sample size was estimated using data from our previous study that a sample size of 12 per group would have >80% power to detect a 20% difference in SV with an alpha error of 0.05 and an effect size of 1.35. Comparisons were made with data from a similar group of 12 (six males and six females) matched controls. The usual practice allows donors weighing 41–50 kg to only donate 350 mL, while those beyond 50 kg could donate either 350 mL or 450 mL on a non-remunerated voluntary basis. Consequently, a variety of permutations could arise from the variability in sex and donation volume to complicate recruitment. Thus, for this study, the control group donors were selected by matching to reduce intergroup variability and also improve the sensitivity of a small sample size comparison. Haemodynamic measurements and psychological assessments were made under the same conditions for the control group, but AMT was not performed.

Statistical analysis was performed by an independent statistician using IBM SPSS Statistics for Windows (Version 22.0, IBM Corp., Armonk, NY, USA). Data was tested for normality in distribution using a Shapiro–Wilk test before selection of an appropriate statistical test. Analysis of variance for repeated measures (ANOVA-RM) was used for intragroup comparison between different time points and paired t-tests were used for intergroup comparison. Findings are presented as mean ± standard deviation (SD). *p* < 0.05 was considered statistically significant.

## 3. Results

Data from 12 donors (height: 166.4 ± 8.8 cm, weight: 61.2 ± 8.9 kg) were analysed and compared with 12 matched control donors (height: 166.5 ± 7.5 cm, weight: 58.7 ± 11.9 kg). No adverse effects from immediate or delayed VVR were reported. Haemodynamic data showing changes throughout the study are summarized in Table 1 and displayed in Figure 1.

### 3.1. Haemodynamic Findings

Baseline haemodynamic measurements of SV, CO and SVR between the two groups were similar; however, the baseline DBP of the AMT group was higher. Compared to baseline, the control group had a significant reduction in SV (11 mL, −16.2%) and CO (1.2 L·min^−1^, −24.5%), with an increase in SVR (273 dyn·s·cm^−5^, +16.2%) during mid-phlebotomy. No haemodynamic changes were observed in the extended AMT group during this period. SV and CO were higher and SVR was lower in the AMT group compared to the control. BP indices of the AMT group were higher at various timepoints throughout the study compared to the control. The CO of the AMT group at the end of the recovery period were also higher compared to the control group (Table 1, Figure 1b).

### 3.2. Questionnaire Data

Pre-donation anxiety scores were similar. Six donors in the AMT group reported post-donation dizziness compared to none in the control. BDRI scores for the immediate and delayed VVR were similar between groups. However, BDRI scores of the AMT group for delayed VVR were lower compared to the scores for immediate VVR (Immediate: 3 ± 3 vs. Delayed: 2 ± 2, *p* = 0.047).

All donors complied with the instructions to perform AMT and self-graded their exertion efforts as 75% of their full capacity. Eleven donors (92%) were willing to use AMT again. Compared to control, donors who performed AMT reported a lower willingness (Control: 88% vs. AMT: 59%, *p* = 0.004) to donate again. Data are summarized in Table 2.

## 4. Discussion

In this study, the effects of performing an extended AMT during blood donation on the haemodynamic changes and VVR symptoms in YFTD were evaluated. Performing AMT during phlebotomy mitigated the haemodynamic disturbances observed in the control group and resulted in improved haemodynamic indices. However, performing AMT was also associated with mild dizziness when compared to the control group. Data from the control donors showed that the most severe haemodynamic disturbance as evidenced by the SV and CO changes, occurred during mid-phlebotomy when 50% of the blood had been collected. Despite losing double this amount by the end of phlebotomy, the haemodynamic indices recovered to almost 90% of their baseline values at 5 min post-donation, as a result of compensatory changes. The haemodynamic indices remained stable thereafter until the end of the study. Although AMT counteracted the haemodynamic disturbance during phlebotomy, findings from the control donors suggest that prophylactically performing AMT during recovery was unnecessary given that the haemodynamic indices recovered spontaneously. Moreover, it may be counterproductive since a number of donors reported dizziness and fewer donors were willing to return and donate again.

AMT is an effective technique for preventing VVR [9,13,14,15] and several mechanisms have been proposed for its action [8,16]. Physiologically, tension in the muscles of the lower body compresses the capacitance vessels to increase venous return to the heart, augmenting CO [17,18]. The action of performing AMT also stimulates the sympathetic system and alleviates fear and anxiety by distracting donors [9]. The haemodynamic findings of this study agree with the findings of Ditto et al. [8] who measured haemodynamic changes using impedance cardiography, and reported higher readings of SV, CO and heart rate (HR) with a lower SVR in donors who performed AMT. Compared to controls, donors who performed AMT in Ditto’s [8] study reported fewer VVR symptoms (14% vs. 39%) and felt less likely to faint (0% vs. 7%).

Unexpectedly, donors in this study who performed AMT during recovery reported more immediate VVR symptoms, with a remarkably higher incidence of dizziness compared to controls. The reason behind this paradoxical finding is unclear. Since the BDRI questionnaires were completed in the refreshment area after donors had mobilized, it is possible that the symptoms were a consequence of orthosis and cerebral hypoperfusion. It can be speculated that by performing AMT for a longer duration, the raised venous pressure may impede compensatory shift of interstitial fluid into the circulation that physiologically compensates for the intravascular loss [19]. It is also possible that the extra muscle work of performing an extended AMT increased the release of local metabolites such as lactic acid, carbon dioxide, potassium, and adenosine [20], which caused vasodilatation in the lower limbs to aggravate orthosis.

AMT is a generic term encompassing many variations of muscle groups used, with or without counter pressures, over an equally variable number of cycles and duration [16]. It was first described for the management of postural hypotension or phobias that result in vasovagal syncope [21,22]. The most popular version described in the context of blood donation employed a five-second on–off cycle and was originally reported by Ditto [23]. However, a longer on–off cycle is usually employed for mitigating postural hypotension and needle phobia in non-donors. A field test was performed to evaluate several techniques previously, and it was found that a longer duration of on-off cycle was more user-friendly, and just as effective for augmenting CO. This version improved compliance especially since AMT had to be performed over a longer duration in this study. It is also the technique that the blood collection agency advises donors to use for mitigating symptoms of delayed VVR, since it is easier to squat down while maintaining muscle tension with a longer cycle, should symptoms worsen. This may explain the lower BDRI scores for delayed VVR in the AMT group.

Although haemodynamic compromise is an important aetiology for VVR, very few studies have measured the effects from interventions such as AMT on the haemodynamic indices such as SV, CO and SVR. As demonstrated in this study, BP measurements were insensitive and remained relatively unchanged despite circulatory depletions that resulted in a 25% decrease in CO during phlebotomy. Haemodynamic indices were measured non-invasively using USCOM, an ultrasonic CO monitor which provided accurate measurements that have been validated in clinical [24,25] and animal [26,27] studies. The sensitivity of measurements provided by this monitor has enabled the detection of small haemodynamic changes in parturients [28] and normal subjects [29,30] resulting from changes in the angle of table tilts. To the best of our knowledge, this is the first description of the use of USCOM for measuring haemodynamic changes during blood donation, and the effects of AMT in YFTD. The measurements in the control group showed the time course for haemodynamic changes during phlebotomy and recovery which were previously unknown. The haemodynamic indices recovered to ≥90% of the baseline values during recovery, probably as a result of compensatory mobilization of blood from the splenic circulation and fluid shift from the interstitial space. Systolic or diastolic BPs of the AMT group were higher than control at various points throughout the study (Figure 1e). Interestingly, baseline diastolic BP of the AMT group was higher, and this may be due to the lingering effect from learning and practicing AMT before commencement of the study.

Several limitations exist in this study. In retrospect, additional haemodynamic measurements should have been taken after donors mobilized. This would have enabled us to ascertain if the dizziness experienced after performing AMT during recovery were haemodynamically related. Another limitation was the small sample size, and donors were not randomized but physically matched to reduce intergroup variability and facilitate better comparison. It is possible that the two groups of donors were different to account for the findings. However, this is unlikely because of the similarity in findings from intergroup comparison that concurs with findings from the intragroup comparison of haemodynamic changes from baseline. The purpose of this pilot study was to facilitate the design of a large-scale outcome study. Nevertheless, the unexpected findings from this study are important. It will be useful for other blood donation services to be aware of this potential issue. To further evaluate the effectiveness of performing AMT after donation to reduce the incidence of delayed VVR, a larger sample size will be needed.

## 5. Conclusions

Although an extended AMT in recovery may mitigate potential haemodynamic disturbances and improve haemodynamic stability, this was associated with a high incidence of dizziness and reduced willingness to donate again. Haemodynamic profiles from this study suggest that AMT should be performed during phlebotomy, but is not necessary during recovery.

## Figures and Tables

**Figure 1 ijerph-18-10541-f001:**
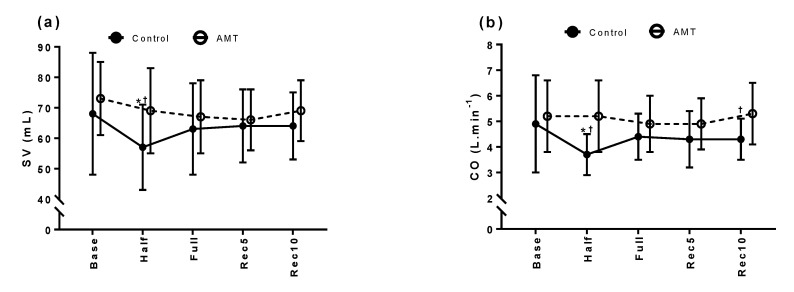
(**a**) Stroke volume (SV), (**b**) cardiac output (CO), (**c**) systemic vascular resistance (SVR), (**d**) heart rate (HR) and (**e**) systolic and diastolic blood pressure (SBP and DBP) during blood donation for control and AMT groups. Data points are presented as mean ± SD. Base = baseline; Half = 50% donated; Full = 100% donated; Rec5 = after 5 min in recovery; Rec10 = after 10 min in recovery. * denotes significant difference from baseline (*p* < 0.05) and † denotes significant difference between groups (*p* < 0.05).

**Table 1 ijerph-18-10541-t001:** Haemodynamic indices and blood pressure during blood donation.

	Group	Baseline	Phlebotomy		Recovery	
			50% Donated	100% Donated	5 min	10 min
SV (mL)	Control	68 ± 20	57 ± 14 ***^,^****^†^**	63 ± 15	64 ± 12	64 ± 11
	AMT	73 ± 12	69 ± 14	67 ± 12	66 ± 10	69 ± 10
CO (L·min^−1^)	Control	4.9 ± 1.9	3.7 ± 0.8 ***^,^****^†^**	4.4 ± 0.9	4.3 ± 1.1	4.3 ± 0.8 **^†^**
	AMT	5.2 ± 1.4	5.2 ± 1.4	4.9 ± 1.1	4.9 ± 1.0	5.3 ± 1.2
SVR (dyn·s·cm^−5^)	Control	1689 ± 945	1962 ± 427 ***^,^****^†^**	1666 ± 416	1640 ± 401	1591 ± 338
	AMT	1523 ± 404	1569 ± 415	1653 ± 332	1547 ± 393	1379 ± 239
HR (beat·min^−1^)	Control	71 ± 14	67 ± 11	70 ± 12	68 ± 13	68 ± 11
	AMT	70 ± 11	75 ± 10	73 ± 8	74 ± 11	77 ± 12
SBP (mmHg)	Control	113 ± 11	116 ± 14	116 ± 11 **^†^**	112 ± 10	110 ± 10 **^†^**
	AMT	118 ± 8	125 ± 11	127 ± 14	118 ± 14	119 ± 10
DBP (mmHg)	Control	72 ± 7 **^†^**	74 ± 8	72 ± 7 **^†^**	70 ± 6 **^†^**	68 ± 9
	AMT	79 ± 5	80 ± 8	81 ± 11	79 ± 12	74 ± 9

Values are presented as mean ± SD. SV = stroke volume; CO = cardiac output; HR = heart rate; SBP = systolic blood pressure; DBP = diastolic blood pressure; SVR = systemic vascular resistance. ***** denotes significant difference from baseline (*p* < 0.05) and **^†^** denotes significant difference between groups (*p* < 0.05). Post hoc analysis demonstrated no statistically significance.

**Table 2 ijerph-18-10541-t002:** STAI Y1, STAI Y2, BDRI (immediate and delayed) scores and willingness to return for Control and AMT groups.

	STAI Y1	STAI Y2	Immediate BDRI	Delayed BDRI	Return Rates
Control	31 ± 8	35 ± 8	1 ± 1	4 ± 10	88% ^†^
AMT	38 ± 11	42 ± 12	3 ± 3	2 ± 2 *	59%

Scores are presented as mean ± SD. STAI = State-Trait Anxiety Inventory; BDRI = Blood Donation Reactions Inventory; Return rates = willingness to donate again. * denotes significant difference between immediate and delayed BDRI in the AMT group (*p* < 0.05) and ^†^ denotes significant difference between groups (*p* < 0.05).

## Data Availability

The data presented in this study are available on request from the corresponding authors.

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
