# Peer review of "Effects of Performing Applied Muscle Tension during Recovery after Phlebotomy in Young, First-Time Donors: A Pilot Study"

_ijerph, 2021, doi:10.3390/ijerph181910541_

Round 1

Reviewer 1 Report

Although the scientific subject is of interest to a certain number of scientists, I do not consider both the number of subjects studied and the methods used, especially with regard to the measurement of hemodynamics, to be suitable for obtaining a greater measure of new scientific knowledge in this field than has been known to date.

Reviewer 2 Report

The paper is interesting however there are editorial errors and statistical explanations are required.

major comments

  • L3 - add information about the pilot study in the title.
  • Introduction - suggests adding a paragraph with statistical information about the need for blood, the number of donors in your country, etc.
  • L100 - 102 - why were these tests used? has the distribution been calculated? if so, write in the manuscript what tests were used to calculate the distribution
    Please add the effect size.

minor comments

  • L4-5 - write surenames with a capital letter and the rest of the letters should be lowercase.
  • L7 /8/10 - add zip codes, and email addresses. (See https://www.mdpi.com/journal/ijerph/instructions  -  mpdi  Microsoft Word template)
  • L43 - "AMT" - expand the abbreviation when you first use it.
  • L45 - do not use spaces between items in quotations. For example, replace" [5, 6, 7] "with "[5,6,7]". Note applies throughout the text.
  • L72 - State-Trait Anxiety Inventory (STAI) - add a citation about the language validation of this questionnaire
  • L111-114 - Values are presented (...) between groups (p<0.05) - remove this information from the table title. Place them under the table.
  • L115-119 - Move this information below the figures. (See https://www.mdpi.com/journal/ijerph/instructions  -  mpdi  Microsoft Word template)
  • L126-127 - remove double interline
  • L144-146 - remove this information from the table title. Place them under the table. (See https://www.mdpi.com/journal/ijerph/instructions  -  mpdi  Microsoft Word template)
  • L247-253 - useauthors' initialsinsteadoffullnames.
  • L276/278/294 - remove bold

Reviewer 3 Report

Overall this is a nice pilot study by Cheung et al. It is novel and the findings are valuable in a blood donation setting and perhaps a vaccination setting too where vaso-vagal syncope is not uncommon. There are some minor aspects to the work that can be improved but it is otherwise suitable for publication.

Minor comments:

62 – which revision of declaration of Helsinki?

73 – are there validation data for this questionnaire? If so, please reference

83 – this sentence (2)  does not make sense – do you mean high/low rather than (yes/no)?

88-89 power calc – is there a reference for this?

94 – do you mean sex rather than gender?

Fig 1 – why are the control and AMT symbols not aligned vertically if they are taken at the same time-points?

Figs – The choice of symbols and dashed lines are confusing. The dash line would work better with hollow symbols and solid lines with the black symbols. The writing needs to be a little larger too in order to increase clarity. I would perhaps remove the small writing with percentages on too – I don’t feel this adds much in my humble opinion.

Panel E on fig 1 – y axis needs more unit markers on the y axis e.g. every 10mmHg

134-135 – need to show the data and stats for this

135-136 – as above

203-205 – you say there was no difference in BP but were BP values not higher throughout in the AMT condition?

Reviewer 4 Report

Authors:

Dear authors,

Thank you very much for the opportunity of reviewing this interesting study with the aim of evaluating the effects of performing an extended applied muscle tension (AMT). Therefore, the practical applications of the study are very interesting. Nevertheless, this study includes some important concerns that must be solved. The most important are the next:

  • Author have selected a sample formed by 12 participants based on a previous estimation. Nevertheless, the sample size includes participants of both sexes. Therefore, authors have included two samples of 6 participants because, how authors consider that don´t exist any difference based on sexes?

It´s very important to analyse the response of the total group, female and male groups. For this, it´s important to perform an ANOVA-RM with the inter-subject factor sex. In addition, it´s essential to include information related to possible sex differences during the text (introduction and discussion section).

  • Why authors didn´t include tests for checking normality and variance of the variables? Please, do it.
  • After applying an ANOVA, it´s essential to perform a Post-Hoc because it´s necessary to detect possible differences between every time points (not only the comparison with baseline values).
  • It´s necessary to report effect size (ES).
  • Based on the little sample size and the limitations included on the text, it´s necessary to title this study as “pilot study” and be more cautelous in the conclusions.
  • Physiological mechanism of AMT must be extended on the introduction section. In addition, it could be very interesting to add information relative to the prevalence of this manoeuvre between donors.

In addition, I would like to indicate other concerns that include the next:

  • The first paragraph of the discussion section must be reduced because is very long and include redundant information of the results section.
  • It must be avoided to use “we” and “our”. Text must be impersonal. Please, to revise it along the document.
  • AMT must be defined the first time that it appears in the text (line 43).
  • The first sentence of the conclusions must be removed.

Round 2

Reviewer 1 Report

I have decided to reject the paper in my first review process.

Author Response

We have received and noted the comments from Reviewer 1

Reviewer 2 Report

The corrections made by the authors are acceptable.

my final comments:
1. suggests adding the effect size to each statistically significant result and adding the exact p-value to each significant result;

2. unify the font in the first reference (L275);

3. remove gaps between references (L276-278/283-285/L297-298).
